# Recent Updates of Diagnosis, Pathophysiology, and Treatment on Osteoarthritis of the Knee

**DOI:** 10.3390/ijms22052619

**Published:** 2021-03-05

**Authors:** Sunhee Jang, Kijun Lee, Ji Hyeon Ju

**Affiliations:** 1Catholic iPSC Research Center, College of Medicine, The Catholic University of Korea, Seoul 06591, Korea; sunshinerosa@naver.com (S.J.); bluebusker@gmail.com (K.L.); 2Division of Rheumatology, Department of Internal Medicine, Seoul St. Mary’s Hospital, College of Medicine, The Catholic University of Korea, Seoul 06591, Korea

**Keywords:** osteoarthritis, diagnosis, management, surgery, cell therapy, embryonic stem cells, induced pluripotent stem cells, mesenchymal stem cells

## Abstract

Osteoarthritis (OA) is a degenerative and chronic joint disease characterized by clinical symptoms and distortion of joint tissues. It primarily damages joint cartilage, causing pain, swelling, and stiffness around the joint. It is the major cause of disability and pain. The prevalence of OA is expected to increase gradually with the aging population and increasing prevalence of obesity. Many potential therapeutic advances have been made in recent years due to the improved understanding of the underlying mechanisms, diagnosis, and management of OA. Embryonic stem cells and induced pluripotent stem cells differentiate into chondrocytes or mesenchymal stem cells (MSCs) and can be used as a source of injectable treatments in the OA joint cavity. MSCs are known to be the most studied cell therapy products in cell-based OA therapy owing to their ability to differentiate into chondrocytes and their immunomodulatory properties. They have the potential to improve cartilage recovery and ultimately restore healthy joints. However, despite currently available therapies and advances in research, unfulfilled medical needs persist for OA treatment. In this review, we focused on the contents of non-cellular and cellular therapies for OA, and briefly summarized the results of clinical trials for cell-based OA therapy to lay a solid application basis for clinical research.

## 1. Introduction

Osteoarthritis (OA) is the most common chronic articular disease and remains one of the few chronic aging disorders with few effective treatments, none of which have been proven to delay disease progression. It can affect small, medium, and large joints, although in terms of a painful disease, the knee is most frequently affected in up to 10% of men and 13% of women aged above 60 years, with evidence of symptomatic OA of the knee in the United States [1].

OA can be defined pathologically, radiographically, and clinically. The most common method for radiographic definition is the Kellgren–Lawrence (KL) radiographic grading system and atlas, which has been used for more than 40 years. This overall joint scoring system grades OA into five levels from 0 to 4, defining OA by the presence of a definite osteophyte (Grade ≥ 2), and more severe grades by the presumed successive appearance of joint space narrowing, sclerosis, cysts, and deformity [1,2]. However, all patients with radiographic OA do not have a clinical condition, and all patients with joint symptoms do not demonstrate radiographic OA [1]. Therefore, OA must be diagnosed using a variety of pathological, clinical, and radiological methods [3].

## 2. Osteoarthritis of the Knee

The knee is the largest synovial joint in humans and consists of bone structures (distal femur, proximal tibia, and patella); cartilage (meniscus and free cartilage); ligaments; infrapatellar fat pad; and synovium. The synovium is responsible for the production of synovial fluid that lubricates and nourishes the vascular cartilage. However, considering the frequent use and high stress on this joint, it is a frequent site of painful conditions, particularly OA [4,5]. Disease evaluation of OA is generally slow and can take years. Successively, the disease can also go through stages or show gradual evolution over time, making the severity and symptoms of the disease worse [6].

## 3. Mechanisms/Pathophysiology

The diarthrodial joint connects two adjacent bones, covered with a special articular cartilage layer, and wrapped in a synovial bursa [7]. The articular cartilage is composed of water (>70%) and organic extracellular matrix components, mainly type II collagen, aggrecan, or other proteoglycans [6]. Chondrocytes detect mechanical stress and changes in the pericellular matrix primarily through receptors on the components of the extracellular matrix. The change in response to mechanical or inflammatory stimulation results in the upregulation of aggrecanase and collagenase. Moreover, receptors on resting chondrocytes are protected from interacting with certain matrix components by the unique composition of the pericellular matrix. Type II collagen-containing networks in the interregional regions are generally not degraded as they are coated with proteoglycans [8].

The main cartilage matrix-degrading enzymes are zinc-dependent metalloproteinases (MMPs) belonging to the MMP and A Disintegrin and Metalloproteinase with Thrombospondin motifs (ADAMTS) families. MMPs include the collagenases MMP-1 and MMP-13 (highly efficient against type II collagen as a substrate), MMP-3 (a potent aggrecanase), and MMP activator [9].

The importance of proteoglycan depletion in cartilage erosion has been demonstrated in ADAMT5 (the primary aggrecanase) knockout mice protected from progression using a surgical OA model [10]. However, aggrecan depletion, by itself, does not drive OA progression, as suggested by studies wherein MMP13 knockout mice exhibited inhibition of cartilage erosion; however, it does not prevent aggrecan depletion [11]. When the collagen network begins to break down, irreversible cartilage breakdown proceeds [8]. Partly, overloading and inflammation leading to cartilage degradation can cause OA. Prostaglandin E2 is one of the major catabolic factors associated with OA, where MMP is crucial for cartilage degeneration. Thus, the mechanosensitive microsomal prostaglandin E synthase type 1 enzyme represents a potential therapeutic target in OA [12].

In OA, there is a gradual disappearance of cartilage associated with chondrocyte loss and phenotypic transformation, including cluster formation and activation of catabolic phenotypic and hypertrophic differentiation. Remodeling of subchondral bone occurs with the development of blood vessels located in structures (vascular channels) that contain osteoblasts and sensory nerves. Vascular channels should facilitate biochemical communication between the bone and cartilage. In response to multiple stimulations, chondrocytes modify the phenotype and express a subset of factors (such as cytokines, chemokines, alarmins, damage-associated molecular pattern, and adipokines). All these mediators act as paracrine factors, begin a vicious cycle of cartilage breakdown, reach the synovial fluid, and trigger an inflammatory process with the production of synovial macrophages and fibroblasts of the factor [8]. Vascular channels have sensory nerve terminations, and the associated innervation of articular cartilage may contribute to tibiofemoral pain in OA across a wide range of structural disease severities [13]. Figure 1 illustrates the associated mechanism [6,8].

The infrapatellar fat pad (IFP) is in close contact with the synovial membrane, and due to the metabolic properties of adipose tissue, IFP may affect the functioning of the synovial membrane [14]. In a recent study, it was found that IFP in OA patients of the knee was more inflamed and vascularized compared to the IFP in anterior cruciate ligament reconstruction patients [15]. It has been recognized that IFP secretes adipocytokines and has more inflammation and fibrotic changes than control group [15]. All of these studies could support the new idea that the IFP and the synovial membrane could be considered a morpho-functional unit [14].

Cartilage is a highly specialized connective tissue, and its damage is the main feature of OA. The primary determinants of OA are namely, aging, genetic predisposition, metabolic syndrome, or trauma, and activation of the inflammatory pathways occurs in cartilage [16]. Chondrocytes form a vicious cycle leading to the progression of OA by producing inflammatory mediators that can lead to cartilage damage and changes in adjacent joint tissue. Therefore, the components of inflammatory pathways to discover disease-modifying OA drugs should be known [8].

## 4. Diagnosis

Although OA is an extremely common illness, its diagnosis may be difficult. Diagnostic criteria were developed for OA of the knee. The primary goal of the diagnostic criteria is to differentiate OA from other arthritis, such as rheumatoid arthritis and ankylosing spondylitis [6].

The American College of Rheumatology (ACR) classification criteria for OA of the knee were used widely [17]. One study demonstrated that crepitus is specific for patellofemoral joint OA rather than tibiofemoral joint OA, as suggested by the magnetic resonance imaging definition [18]. Another arthroscopy-based study reported an association between crepitus and cartilage pathology in both compartments of the knee [19].

Cartilage degeneration and other skeletal changes can be examined radiographically and quantified using the semi-quantitative grading scale known as the KL scale [20], Ahlbäck classification [21], and knee osteoarthritis grading system (KOGS) [22]. The original definitions of the KL scale, Ahlbäck classification, and KOGS are shown in Table 1. A KL grade picture is shown in Figure 2 [21].

## 5. Risk Factor

Previous knee trauma increases the risk of osteoarthritis of the knee by 3.86 times [23]. Old age, female, overweight and obesity, repetitive use of joints, bone density, muscle weakness, and joint relaxation all play an important role in the development of knee OA [23]. Also, frequent squatting is a risk factor for knee OA [23].

## 6. Current Point-of-Care Treatment

### Conventional Management

Currently, various guidelines have been developed to standardize and recommend available treatments by academic and professional societies. Table 2 shows the available treatment options from the Osteoarthritis Research Society International (OARSI), ACR, and the American Academy of Orthopedic Surgeons (AAOS) publications [24,25,26].

## 7. Interventional Management

Multiple substances that are delivered through intra-articular (IA) injections have been explored. The idea is that local treatments (IA injection) will have less systemic side effects and placing the drug inside the joint will have a more direct effect. Studies have shown that IA therapy is more effective than oral non-steroidal anti-inflammatory drugs and other systemic pharmacological treatments; however, it has also revealed that some of its benefits may be secondary to the IA placebo effect [27,28].

Also, injectable drug delivery alone, including new treatments, may not provide significant benefits for OA treatment. Because IA injections cannot target the complexity of the pathological mechanisms [29]. It means that relatively new concept is the multimodal approach to the IA injections, which is needed to significant effect on the entire knee.

### 7.1. Intra-Articular Corticosteroid Injection

In the knee joint of OA, IA corticosteroid injections are usually conditionally recommended over other forms of IA injection. There are few one-to-one comparisons; however, the evidence for the efficacy of glucocorticoid injections is significantly higher than that of other drugs [25]. Corticoids act directly on nuclear receptors, disrupting the inflammatory cascade at several levels, causing immunosuppressive and anti-inflammatory effects. They are part of the pain relief mechanism and reduce the action and production of interleukin-1, prostaglandins, leukotriene, MMP9, and MMP-11, which are believed to increase joint mobility in the knee OA [4]. The clinical anti-inflammatory effects of these actions include decrease in erythema, heat, swelling, and tenderness of the inflammatory joints, and increases in relative viscosity with an increase in hyaluronic acid (HA) concentration [28]. Therefore, IA corticosteroid injections reduce acute pain episodes and increase joint mobility, particularly when there is evidence of inflammation and joint effusion during OA redness [30].

The current Food and Drug Administration approved immediate release corticosteroids for IA usages namely, methylprednisolone acetate, triamcinolone acetate, triamcinolone hexacetonide, betamethasone acetate, betamethasone sodium phosphate, and dexamethasone [28].

In summary, research evidence shows that IA corticosteroid injections provide a short-term reduction in OA pain and act as an adjunct to key therapy for moderate to severe pain relief in patients with OA [31].

### 7.2. Intra-Articular Hyaluronic Acid Injection (Viscosupplementation)

HA, a viscoelastic mucopolysaccharide component of synovial fluid, is produced from harvested rooster combs or through in vitro bacterial fermentation [32]. HA is a high-molecular-weight glycosaminoglycan that consists of a repeating sequence of disaccharide units composed of N-acetyl glucosamine and glucuronic acid [33]. Viscous supplementation through an IA injection of HA is aimed at restoring the beneficial environment present in non-arthritic joints. Additionally, the safety profile of such injections for painful knee OA is well established [32].

Previous studies demonstrated obvious benefits of intra-articular HA injection; however, according to the 2019 ACR/European League Against Rheumatism study, the benefit was restricted to studies with a higher risk of bias compared to saline injection. Therefore, in recent years, HA injection has been conditionally recommended to control joint symptoms when glucocorticoid injection or other interventions fail [25].

### 7.3. Intra-Articular Platelet-Rich Plasma Injection

The IA platelet-rich plasma (PRP) injection has emerged as a good treatment for knee OA. Several randomized controlled trials have been shown that PRP is a safe and effective treatment. At this time, IA PRP is not a standard treatment of the knee OA, but it is similar in efficacy to HA, and appears to be more effective than HA in young, active patients with low-grade OA [34].

## 8. Surgery

The goals of surgery for patients with OA are to reduce pain, minimize disability, and improve quality of life. Treatment should be individualized according to the functional condition of the patients, severity of the disease, and nature of the underlying disease. Surgical intervention for patients with OA is generally performed when a less invasive treatment is unsuccessful.

### 8.1. Total Knee Replacement Surgery (Total Knee Arthroplasty)

Total knee replacement (TKR) surgery involves excising the damaged ends of the tibia and femur and capping both using a prosthesis. Both prostheses comprise durable plastic. These new surfaces move smoothly with each other. Partial recovery takes 6 weeks and complete recovery takes up to 1 year [35].

A randomized, controlled trial of TKR demonstrated that non-surgical treatment after TKR is superior to non-surgical treatment alone in providing pain relief in patients with moderate to severe OA of the knee and improving function and quality of life after 1 year. However, clinically relevant improvements were seen in both groups, and patients who received TKR had more severe side effects [36].

In addition, IFP resection during TKR is the subject of an ongoing debate without clear consensus [37].

### 8.2. Partial Knee Replacement Surgery (Unicompartmental Knee Arthroplasty)

Unicompartmental knee arthroplasty is an alternative to TKR for patients whose disease is limited to a single area of the knee, particularly the isolated tibiofemoral compartment (medial or lateral). As partial knee replacement is performed using smaller incisions, patients can generally be discharged earlier than those who undergo TKR and can return sooner to normal activities, including work and sports [38].

### 8.3. Knee Osteotomy (High Tibial Osteotomy or Femoral Osteotomy)

High tibial osteotomy is a surgery to realign the knee joint. It is more important for the treatment of cartilage damage or OA of the medial compartment with varus deformity. High tibial osteotomy creates a postoperative valgus limb alignment by lateral movement of the load-bearing axis of the lower limb [39].

### 8.4. Knee Arthroscopy

Knee arthroscopy is most commonly performed to treat OA or meniscus problems. An arthroscopy requires a small incision in the skin with the insertion of a camera on a stick. Another incision is needed to insert other instruments and treat the disease [40].

### 8.5. Knee Cartilage Repair and Cartilage Restoration

Many surgical techniques have been developed to address focal cartilage defects. Cartilage surgery strategies include palliative (chondroplasty and debris removal); repair (perforation and microfracture); or restoration (auto chondral cell transplant, osteochondral autograft, and bone cartilage allograft) [41].

## 9. Cellular & Experimental Therapy

### 9.1. Cellular Therapy

Several cell therapeutic attempts have been made to regenerate damaged joint cartilage. Autologous chondrocyte implantation (ACI) has been proposed as a surgical technique for partial cartilage lesions [42]. ACI is known as the most traditional cell-based therapy that has evolved with a high success rate. However, since it is limited to the damaged cartilage area, it is difficult to use in general OA treatment [43,44,45]. Mechanical, biological, and chemical scaffold-based approaches have also been developed to allow autologous chondrocytes to fill the cartilage lesions. With the scaffold, chondrocytes are less prone to dedifferentiation, and more favorable cartilage can be produced [46]. However, the limited number of primary chondrocytes has shown therapeutic limitations. Consequently, stem cell-based therapies have been developed to compensate for these shortcomings (Figure 3). Stem cells are undifferentiated cells capable of differentiating into various specialized cells such as bone cells, chondrocytes, and adipocytes [47]. Additionally, stem cells are characterized by the ability to release cytokine secretions that can downregulate several important inflammations [48]. Three other types of stem cells, embryonic stem cells (ESCs), induced pluripotent stem cells (iPSCs), and mesenchymal stem cells (MSCs) are also potential candidates for cartilage regeneration for the OA treatment. Both ESCs and iPSCs have intrinsically pluripotent features and can differentiate into other cell types, including chondrocytes. Several studies have shown that chondrocytes can be differentiated using ESCs and iPSCs [49,50]. However, there is a risk that both stem cells can form teratoma and immunogenicity [51].

Mesenchymal stem cells (MSCs) are not as pluripotent as ESCs and iPSCs; however, may be considered the most ideal among the various types of stem cells for OA treatment [52]. MSCs have advantages that can be obtained in various ways such as bone marrow [53], adipose tissue [54], and umbilical cord [55]. As MSCs express and secrete various growth factors and cytokines and have anti-inflammatory activity, numerous studies have been conducted for the treatment of OA [56,57,58]. In addition to these sources of stem cells, infrapatellar fat pad (IFP) have been recently considered as a source of stem cells for cartilage regeneration in OA due to their increased chondrogenic capacity [59,60]. However, IFP-derived stem cells seem to be primed by the pathological environment and exert a protective role in the inflammatory environment [60]. Thus, further research is needed to clarify this point.

Human ESCs can be used as a raw material for cell therapy for the treatment of OA. It has been demonstrated that MSCs differentiated from ESCs have similar efficacy to those extracted from somatic tissues, such as bone marrow, and can treat various autoimmune and inflammatory diseases [61,62,63]. ESC-derived MSCs are more advantageous for use as a cell therapy than natural MSCs extracted from bone marrow tissues (Table 3). ESCs can be produced on a large scale from human ESC raw materials that can be supplied endlessly.

Additionally, human ESCs may be a potential treatment option as a source of consistently homogeneous cells with high chondrogenic ability [49]. Sources of expandable cartilage precursors may have broad potential to advance articular cartilage therapy and disease modeling and act as therapeutics that can promote cartilage regeneration or prevent degeneration [64,65]. Although several studies have attempted to treat OA using specialized cells derived from human ESCs [66,67,68], clinical studies on effective human ESC-derived cell therapeutics in patients with OA are still needed.

### 9.2. Induced Pluripotent Stem Cells for Osteoarthritis

The general differences in ESCs and iPSCs and the use of human PSCs for disease modeling have been extensively discussed in the literature [69,70,71]. The development of iPSCs opens up new horizons for the development of new research tools for OA that do not yet have a clear treatment [72]. The iPSCs, reprogrammed from somatic cells [73,74], provided a new opportunity to create a virtually unlimited number of patient-specific stem cells for OA for drug discovery. Thus, chondrogenic differentiation of iPSCs from patients with symptoms of OA may enable many studies of cartilage tissue [42,50,75]. Clinical studies using iPSCs in cell therapy for OA are still in the basic stage with an understanding of the cartilage regeneration mechanisms. iPSCs have proliferative and differentiation capabilities similar to those of other stem cells; however, do not have immune rejection reactions and ethical problems [76]. Additionally, studies on a new method of producing iPSCs without the use of viral vectors have been actively conducted in recent years to reduce the risk of tumorigenicity [77,78]. Nevertheless, there is still limited data on the effects of iPSCs on cartilage formation and OA, and further studies are needed (Table 4).

### 9.3. Mesenchymal Stem Cells for Osteoarthritis

MSCs are pluripotent progenitor cells derived from a population of adult stem cells that can be isolated from numerous tissues, including bone marrow, peripheral blood, adipose tissue, synovium, placenta, and umbilical cord [79,80]. Human MSCs are defined as cells that adhere to plastics; are positive for CD105, CD73 cell surface markers; negative for CD45, CD34 cell surface markers; and differentiate into osteoblasts, chondrocytes, and adipocytes [81]. Additionally, MSCs have unique immunomodulatory properties and can reduce inflammation and support other cells, enhancing angiogenesis, cell survival, and differentiation [82,83].

The primary isolated stromal cell represents the best option for OA treatment [84]. Bone marrow-derived stromal cells are the most common clinical source of MSC [85,86]. Although the main source has been on the use of bone marrow-derived stromal cells, some researchers have chosen to use adipose tissue-derived stromal cells as an alternative cell line [87,88]. These are harvested from bone marrow concentrate containing hematopoietic stem cells, endothelial progenitor cells, and related cytokines and growth factors [89]. Currently, translational medical research targeting MSCs for OA in the clinical trial database is promising. Clinical trials using MSCs in knee OA are actively underway. However, careful evaluation of clinical outcome data is necessary. The results appear to focus primarily on safety and efficacy [90]. Several studies have reported clinical trials for IA injection of MSCs in patients with OAs (Table 4) [91,92,93,94,95,96,97,98,99,100,101].

**Table 4 ijms-22-02619-t004:** Detailed clinical studies of ESCs, iPSCs, and MSCs in OA.

Cell Source	No. of Participants	Mean Follow-Up (Months)	Delivery Methods	Clinical Outcome	Reference/NCT
ESCs	N/A	N/A	N/A	N/A	N/A
iPSCs	N/A	N/A	N/A	N/A	N/A
BMSCs	45	75	Two-stage surgical approaches	No risk of serious complications	[91] N/A
BMSCs	4	12	IA injection	Improved pain, walking, and stairs climbing	[92] 00550524
BMSCs	56	24	IA injection	Better clinical outcomes and MRI in MSCs group	[93] N/A
BMSCs	12	12	IA injection	Improvement of cartilage quality on MRI	[94,95] 03956719
BMSCs	3	60	IA injection	Better than the baseline level	[96] 00550524
AMSCs (ASF)	18	6	IA injection	Better clinical outcomes	[97] 01300598
AMSCs (ASF)	100	26	IA injection	Improved pain VAS scores	[98] N/A
AMSCs (GSF)	40	29	IA injection and surgical implantation	Improved clinical outcomes	[99] N/A
AMSCs (ASF)	18	6	IA injection	Improved clinical outcomes	[100] 01585857
AMSCs (ASF)	24	6	IA injection	Improved clinical outcomes	[101] 02658344

ESCs; embryonic stem cells, iPSCs; induced pluripotent stem cells, BMSCs; bone-marrow-derived mesenchymal stem cells, AMSCs; adipose-derived mesenchymal stem cells, ASF; abdominal subcutaneous fat, GSF; gluteal subcutaneous fat, IA injection; intra-articular injection, MRI; magnetic resonance imaging, VAS; visual analog scale, NCT; The national clinical trial number, N/A; Not Assigned.

It has been suggested that the secretion of trophic factors, where exosomes play an important role, contributes to the MSC-based therapeutic mechanism of OA [102,103]. The paracrine secretion of MSC-derived exosomes may play a role in the repair of joint tissue as well as MSC-based treatments for other disorders. Recent studies have shown that MSC-derived exosomes may inhibit OA development and have summarized findings on exosomes derived from various MSCs and their effectiveness in OA therapy [104,105].

## 10. Noncellular Therapy

### Gene Therapy

Gene therapy with genes encoding cartilage growth factor and anti-inflammatory cytokines is of interest in treating OA. Gene transfer was conducted in two ways: (1) In vivo injection/intravenous administration into the joint; (2) Ex vivo exposition/cell harvesting from the patient to the vector, and returning the modified cells to the joint [106].

In 2017, in vitro TGF-β1 gene therapy using retrovirus was approved for allogeneic chondrocytes [107]. In 2018, Kim et al. reported the clinical efficacy of TissueGene-C (TG-C), a cell and gene therapy for human knee OA consisting of non-transfected and transduced chondrocytes transduced using retrovirus to overexpress TGF-β1. They concluded that TG-C was associated with a statistically significant improvement in function and pain in patients with knee OA [108].

## 11. Conclusions

OA is the most common chronic joint disease, associated with obesity, aging, and socioeconomic impact. The mechanism works complex, local and systemic factors modulate clinical and structural representation, sometimes resulting in a common end-course of joint destruction. Treatment goals are to relieve symptoms and improve quality of life. Conventional management, surgery, and experimental therapy are summarized above.

## Figures and Tables

**Figure 1 ijms-22-02619-f001:**
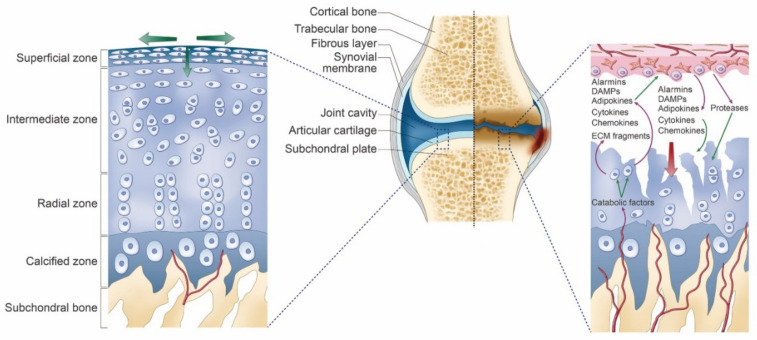
Mechanisms for the osteoarthritis of the knee. Healthy articular cartilage (Left)—Because of absence of vessels within cartilage, chondrocytes can live in a hypoxic environment. Hypoxia is important for chondrocyte function and survival. The main function of cartilage is the absorption and the removal of mechanical load, which is necessary to maintain cartilage homeostasis. Osteoarthritis articular cartilage (Right)—Development of vessels (called vascular channels) are supposed to facilitate biochemical communication between the bone and the cartilage (such as cytokines, chemokines, alarmins). It initiates a vicious cycle of cartilage degradation.

**Figure 2 ijms-22-02619-f002:**
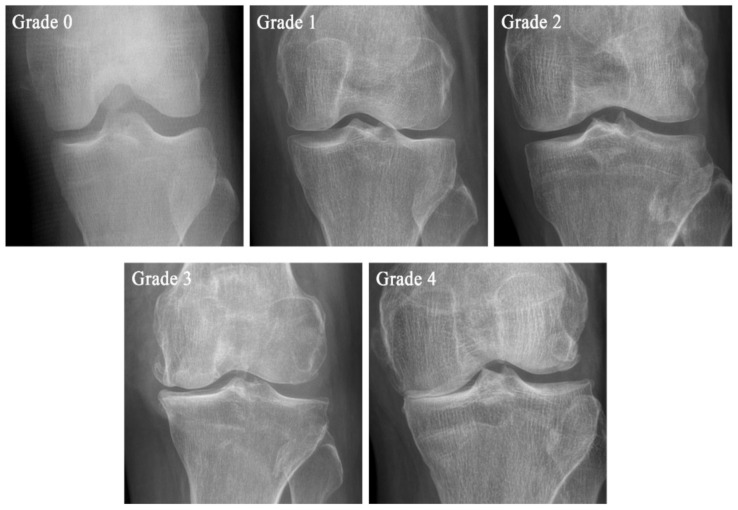
The example of the Kellgren–Lawrence (KL) scale. KL classification is the most widely used radiographic scale. The radiograph was recorded at St. Mary’s Hospital in Seoul.

**Figure 3 ijms-22-02619-f003:**
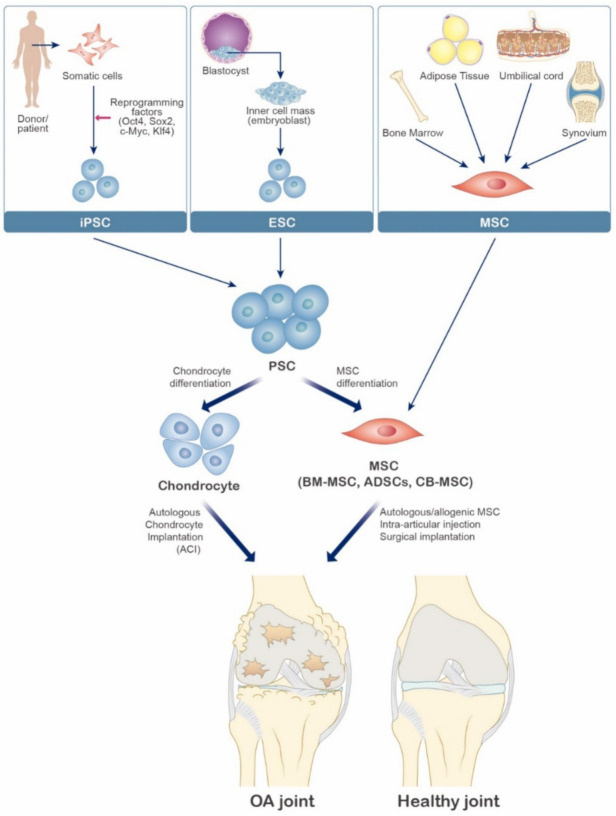
Schematic diagram of various cell-based therapy for osteoarthritis. Embryonic stem cells (ESC), induced pluripotent stem cells (iPSC), and mesenchymal stem cells (MSC) are potential candidates for cartilage regeneration for the OA treatment. MSC can be isolated from bone marrow, adipose tissue, umbilical cord, synovium. Pluripotent stem cells (PSC), including ESC and iPSC, are considered sources for the derivation of chondrocytes and MSC.

**Table 1 ijms-22-02619-t001:** Comparison of the original definitions of the Kellgren–Lawrence (KL) scale, Ahlbäck clasScheme 20.

Grade	KL Scale	Ahlbäck Classification	KOGS
Grade 0	No pathological features of osteoarthritis (OA)		
Grade 1	Suspicious narrowing of the joint space and possible osseous lip	Joint space narrowing, with or without subchondral sclerosis. Joint space narrowing is defined by this system as a joint space <3 mm, or less than half of the space in the other compartment, or less than half of the space of the homologous compartment of the other knee	An isolated medial, lateral tibiofemoral, or patella-femoral joint OA with ligament stability and two functionally intact compartments
Grade 2	Clear bone tissue and possible stenosis of the joint space	Obliteration of the joint space	Deteriorating isolated lesion with ligament stability and a correctible coronal subluxation
Grade 3	Moderate multiple bone tissue, clear narrowing of the joint space, slight sclerosis, and possible deformity of the ends of the bones	Bone defect/loss < 5 mm	Includes an isolated medial or lateral tibiofemoral OA and concomitant pathologies such as anterior cruciate ligament deficiency (3A) or grooving of patella-femoral joint or patellectomy (3B)
Grade 4	Large bone tissue, marked narrowing of the joint space, severe sclerosis, and clear deformities of the ends of the bones	Bone defect/loss between 5 mm and 10 mm	Includes cases of bi-compartmental tibiofemoral OA without concomitant ligament instability (4A) and with ligament instability (4B)
Grade 5		Bone defect/loss >10 mm, often with subluxation and arthritis of the other compartment	

**Table 2 ijms-22-02619-t002:** Osteoarthritis of the knee management recommendations from the three societies.

Treatment	OARSI	ACR	AAOS
Exercise (Land-based)	Appropriate	Strong recommendation	Strong recommendation
Exercise (Water-based)	Appropriate	Strong recommendation	Strong recommendation
Transcutaneous electrical nerve stimulation	Uncertain	Strong recommendation against use	Inconclusive
Cane (Walking stick)	Appropriate	Strong recommendation	
Weight control	Appropriate	Strong recommendation	Moderate recommendation
Chondroitin or Glucosamine	Not appropriate for disease modification, Uncertain (Sx relief)	Strong recommendation against use	Recommendation against use
Acetaminophen	Without comorbidities: appropriate	Conditional recommendation	Inconclusive
Duloxetine	Appropriate	Conditional recommendation	No recommendation
Oral NSAIDs	Without comorbidities: appropriate; With comorbidities: Uncertain	Strong recommendation	Strong recommendation
Topical NSAIDs	Appropriate	Conditional recommendation against use	Strong recommendation
Opioids	Uncertain	No recommendation	Recommended (only tramadol)
Intra-articular corticosteroids	Appropriate	Strong recommendation	Inconclusive
Intra-articular viscosupplementation	Uncertain	Conditional recommendation against use	Recommendation against use

OARSI: Osteoarthritis Research Society International; ACR: American College of Rheumatology; AAOS: American Academy of Orthopedic Surgeons; NSAIDs: non-steroidal anti-inflammatory drugs.

**Table 3 ijms-22-02619-t003:** Summary of advantages and disadvantages of ESCs, iPSCs, and MSCs.

	Advantages	Disadvantages
Embryonic stem cells (ESCs)	Unlimited self-renewal	Ethical concerns
	Unlimited proliferation	Tumorigenic potential
	Pluripotent	Difficulty in vitro work
	Potentially unlimited supply	Difficulty in controlling differentiation
Induced pluripotent stem cells (iPSCs)	Autologous origin	Security
	Extensive sources	Tumorigenic potential
	Unlimited self-renewal	Inefficiency
	Unlimited proliferation	Instability
	Pluripotent	Unclear mechanism
	No ethical issues	Difficulty in controlling differentiation
Mesenchymal stem cells (MSCs)	High chondrogenic potential	A limited number of cells
	Good expansion ability	More affected by donor age
	Easily accessible, reliable for isolation	Donor site pain

## Data Availability

No new data were created or analyzed in this study. Data sharing is not applicable to this article.

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
