# Peer review of "Recent Updates of Diagnosis, Pathophysiology, and Treatment on Osteoarthritis of the Knee"

_ijms, 2021, doi:10.3390/ijms22052619_

Round 1

Reviewer 1 Report

A nicely written review in a timely manner. However, to review the pathophysiology and therapy of the whole spectrum of osteoarthritis in a single paper is a very ambiguous task there are a lot of similar reviews out there and the authors should explain their aim and justify their idea for writing such a paper. Besides, “knee” should be included in the title since the authors are mainly talking about it. Diagnosis and diagnostic approaches are thoroughly represented in the manuscript body with tables/figures/subsections; thus, "diagnosis" or relevant words could be included in the title.

Since your title suggest that the new concepts of OA will be described, there is no need to present the old ACR criteria for knee OA, a single reference should do the work. Furthermore, those criteria are classification and diagnostic (in Table 1).

A relatively new concept is the multimodal approach to the intra-articular injections where more than one modality in addition to injections is needed to significantly affect the knee as a whole (DOI: 10.1007/s00296-020-04681-7).

Interventions on modifiable risk factors (the cornerstone of modern non-surgical therapy) may be included.

Summary ("conclusion" is the preferable term) should be reworked. What does it mean “The concept of mechanism is complex” or “resulting in a common end-course of 327 joint destruction.”.

Overall, the manuscript needs minor English improvement. Please consult with an expert with full proficiency in English.

Author Response

Thank you for your constructive review.
Please see the attachment.

Reviewer 2 Report

The narrative review entitled “Recent Updates of Pathophysiology and Treatment on Osteoarthritis” is an interesting and well-written manuscript.

My major concern is about the fact that the authors focused the attention on the role of cartilage in OA. However, the reader expects to read the recent updates on OA and not just on cartilage and OA. To this regard, the authors leaved out the key role of synovial membrane and did not include the findings on the role of infrapatellar fat pad in the paragraphs osteoarthritis of the knee and mechanisms/pathophysiology. Recently, it has been recognized that this tissue secretes adipocytokines, it is more inflamed and fibrotic compared to controls (doi: 10.3390/ijms21176016). Moreover, understanding the role of this adipose tissue is important also concerning total knee replacement. Indeed, it is still under debate if this tissue should or should not be removed during surgery. Finally, it has been suggested that synovial membrane and infrapatellar fat pad seem to function as anatomo-functional unit. The authors should improve the first part of the review discussing these recent findings.

Lines 46-48: “However, all patients with radiographic OA do not have a clinical condition, and all patients with joint symptoms do not demonstrate radiographic OA.” A reference should be added.

It is not possible to see the entire figure 1. The authors should align this figure.

Line 128-129: it is not clear why the authors reported the following sentence in the text: The radiograph was recorded at St. Mary’s Hospital in Seoul. The authors should move this sentences in figure 2 legend. Moreover, it is not clear why the authors reported only KL in figure 2 and not examples of the other classification systems.

The authors discussed about intra-articular corticosteroids injection and intra-articular viscosupplementation. However, they did not even mention platelet-rich plasma injection.

Line 234: the authors discuss about stem cells and then about iPSC and ESC. Did the authors refer to mesenchymal stem cells at line 234?

The legend of Figure 3 seems to behind the figure. Thus, it is not possible to read it.

Lines 261-262: the authors reported that several studies have attempted to treat OA using specialized cells derived from human ESCs. However, they cited only a study.

Regarding the mesenchymal stem cells paragraph, the authors focused overall on BMSC but in table 5 they reported also clinical studies on AMSCs. Could the authors add the clinical trial number in the table? I suggest to add information about the use of AMSCs also in the text of the paragraph. Moreover, the authors should specify the origin of these AMSCs (subcutaneous or abdominal?). It should also be noted that also infrapatellar fat pad has been suggested to be a potential source of stem cells but it has been also showed as these stem cells might be primed by the pathological OA environment (doi.org/10.3389/fcell.2019.00323; doi.org/10.1155/2017/6843727).

Author Response

(The authors gave the same response as above.)

Round 2

Reviewer 1 Report

All of my concerns were addressed. Language is properly edited. I have no additional comments.

Author Response

Thank you very much for your confirmation.

Reviewer 2 Report

The manuscript improved after the revision. I just noticed that the authors added a brief hint that IFP tissue may also be a source of stem cells (as suggested by me). However, I suggest to specify that, although there are several studies on their possible use, one of them (Stocco et al.) suggests that these cells seem to be primed by the pathological environment. Thus, further research is needed to clarify this point. 

Author Response

Thank you very much for your confirmation.
As you suggested, we added the sentences 'However, IFP-derived stem cells seem to be primed by the pathological environment and exert a protective role in the inflammatory environment. Thus, further research is needed to clarify this point.' (Line 289-291)